# Dietary Aspects to Incorporate in the Creation of a Mobile Image-Based Dietary Assessment Tool to Manage and Improve Diabetes

**DOI:** 10.3390/nu13041179

**Published:** 2021-04-02

**Authors:** Yue Qin, Marah Aqeel, Fengqing Zhu, Edward J. Delp, Heather A. Eicher-Miller

**Affiliations:** 1Department of Nutrition Science, Purdue University, West Lafayette, IN 47907, USA; qin53@purdue.edu (Y.Q.); aqeel@purdue.edu (M.A.); 2School of Electrical and Computer Engineering, Purdue University, West Lafayette, IN 47907, USA; zhu0@purdue.edu (F.Z.); ace@purdue.edu (E.J.D.)

**Keywords:** dietary behavior, time of eating, diabetes management, mobile image-based dietary assessment tool, narrative review

## Abstract

Diabetes is the seventh leading cause of death in United States. Dietary intake and behaviors are essential components of diabetes management. Growing evidence suggests dietary components beyond carbohydrates may critically impact glycemic control. Assessment tools on mobile platforms have the ability to capture multiple aspects of dietary behavior in real-time throughout the day to inform and improve diabetes management and insulin dosing. The objective of this narrative review was to summarize evidence related to dietary behaviors and composition to inform a mobile image-based dietary assessment tool for managing glycemic control of both diabetes types (type 1 and type 2 diabetes). This review investigated the following topics amongst those with diabetes: (1) the role of time of eating occasion on indicators of glycemic control; and (2) the role of macronutrient composition of meals on indicators of glycemic control. A search for articles published after 2000 was completed in PubMed with the following sets of keywords “diabetes/diabetes management/diabetes prevention/diabetes risk”, “dietary behavior/eating patterns/temporal/meal timing/meal frequency”, and “macronutrient composition/glycemic index”. Results showed eating behaviors and meal macronutrient composition may affect glycemic control. Specifically, breakfast skipping, late eating and frequent meal consumption might be associated with poor glycemic control while macronutrient composition and order of the meal could also affect glycemic control. These factors should be considered in designing a dietary assessment tool, which may optimize diabetes management to reduce the burden of this disease.

## 1. Introduction

According to the latest report from the United States Centers for Disease Control and Prevention, 34.2 million people of all ages have diabetes, making diabetes the seventh leading cause of death in the United States (U.S.) [1]. Increased adiposity, poor diet, sedentary lifestyle and genetic factors are primary causes of type 2 diabetes (T2D) and also contribute to difficulty in management of type 1 diabetes (T1D) [2]. There are several components of diabetes management including nutrition therapy, insulin therapy, and other glucose-lowering medications. Although dietary interventions may not eliminate or reverse the need for medication and insulin, this lifestyle modification could reduce the need for these therapies in certain scenarios for patients with diabetes (PWD) (including T1D and T2D) [3,4]. One of the important aspects in dietary interventions that aim to prevent T2D incidence or control complications and delay progression in PWD, has focused on carefully monitoring or restricting carbohydrate intake to improve glycemic control [3]. According to the American Diabetes Association, carbohydrate counting is a practice in which PWD count the grams of carbohydrates in a meal in order to match the amount of insulin dose (most effectively given 30 min pre-meal to be effective when glucose enters the body or post-meal in certain cases) necessary to keep postprandial (PP) plasma glucose (1–2 h after eating) below 180 mg/dL [5,6]. This single-nutrient focused practice also fosters food choice flexibility and variety [7]. When combined with supervision from healthcare professionals (registered dietitian or Certified Diabetes Care and Education Specialist), carbohydrate counting has been used as the gold standard for determining insulin dosing for PWD to maintain glycemic control [8,9]. Yet, carbohydrate counting is challenging; miss-estimation is common [10,11,12], and unfortunately, most PWD in the U.S. do not have access to multi-disciplinary care. Other challenges to carbohydrate counting include the time burden required for recording and calculating food intake and maintaining a stable weight within the dietary flexibility that carbohydrate counting allows [7]. Optimal dosing of insulin based on carbohydrate counting may also be difficult to achieve. For example, intake of other macronutrients including fat or protein may result in hyperglycemia and poor glycemic control when insulin dosing is based solely on carbohydrate counting [13,14,15] as noted in the standard diabetes guidelines of the U.S. and Canada [16,17]. In vitro research has also shown molecular mechanisms of macronutrient interactions which could affect the digestion process and glycemic responses [18]. Research on day-time patterns of carbohydrate intake has found that varying levels of carbohydrate intake (categorized by different percentage of total energy) were associated with different times of eating throughout a day [19]. Limited guidance is provided on meal timing for glycemic control. Developments such as automated and comprehensive dietary assessment tools may improve carbohydrate estimation, capture additional macronutrient information and record time of eating occasions that may be used to inform tailored dosing strategies [20] to achieve optimal PP glycemic control in PWD.

A growing number of studies have examined dietary behaviors that may influence glycemic control, including timing and frequency of food intake. Knowledge of these behaviors presents a need for the development of tools that are capable of capturing and integrating data quantifying these behaviors in real-time and investigating how they may influence insulin dosing. Mobile-based applications have the capacity to fulfill these needs. For example, dietary intake may be quantified using image-based methods. For metabolic conditions like diabetes where lifestyle modification is a key factor in disease management, a mobile image-based dietary assessment tool (MIDAT) may provide accessible and responsive platforms for systematic monitoring of dietary intake and behaviors [21,22], communication with healthcare providers, and tailored insulin dosing interventions [23].

Evidence shows that mobile image-based assessment methods are more accurate than traditional dietary recall for collecting information on food intake and show a potential to improve disease outcomes [20,24]. Mobile-based interventions including assessing dietary intake and monitoring lifestyle factors and receiving feedback have led to decreased glycated hemoglobin (HbA1c) in PWD and better weight control for obesity prevention as a result of improved lifestyle factors compared to those not receiving an intervention [22]. Some examples of mobile technologies for assessing dietary or energy intake are camera-scan-sensor technologies, remote food photography methods, and real-time food recognition systems [24]. These existing MIDATs, built for research or for self-monitoring dietary intake, still have several limitations including their ability to accurately identify and quantify the macro/micronutrient content of foods and beverages from a simple picture as well as their ability to capture eating behaviors (eating time and frequency) and dietary intake variability [24]. Nevertheless, a MIDAT that captures detailed dietary information designed for diabetes management, specifically focusing on improving glycemic control for PWD and connecting to insulin delivery mechanisms, has the potential to achieve personalized, real-time insulin dosing based on users’ dietary behaviors. Currently, however, a gap exists in determining what information should be captured to more comprehensively and accurately link dietary behaviors to individualized health outcome data for monitoring dietary intake, real-time insulin dosing modifications and tailoring weight management strategies. The current review will discuss the relevant dietary factors linked to glycemic control that could be incorporated and recorded in a MIDAT, but currently are not, to make it a better tool for managing glycemic control among those on insulin therapy.

The objective of this narrative review is to summarize evidence from the literature that 1. evaluates the role of time of eating occasions on biomarkers of glycemic control; and 2. examines how the macronutrient composition of meals affects glycemic control to inform the development of a MIDAT that is capable of measuring these potentially important components of dietary intake for glycemic control. Potential mechanisms for findings are also briefly summarized. A final discussion section comprehensively integrates the information and implications relative to the overall objectives of the review.

## 2. Materials and Methods

A search for articles published after 2000 was completed using PubMed. The following sets of keywords were included in the search for articles that addressed the topics proposed in the objectives of this review: “diabetes/diabetes management/diabetes prevention/diabetes risk”, “dietary behavior/eating patterns/temporal/meal timing/meal frequency”, and “macronutrient composition/glycemic index/insulin dosing”. Inclusion criteria specified studies conducted in PWD (either T1D or T2D) and were written in English. Reference lists of both primary and review articles were screened for additional potentially relevant articles. A total of 32 eligible articles were included in the current review.

## 3. Results

### 3.1. Evaluating the Role of Time of Eating Occasion on Biomarkers of Glycemic Control in Patients with Diabetes (PWD)

Daily behaviors of dietary intake such as the time of eating, breakfast consumption, evening/late eating habits and meal frequency may play an important role in regulating internal circadian rhythms, which could affect blood glucose control [25]. The time of eating may be associated with a number of cardiovascular biomarkers [26,27]. MIDATs are capable of providing a timestamp on collected images and other data input and may potentially serve as novel and powerful tools in diabetes management by integrating image-based dietary intake with insulin dosing in real-time, empowering PWD to enhance their glycemic control. Current evidence examining the association between time of eating with health indicators (HbA1c, fasting and PP blood glucose levels and glucose area under curve, AUC) in PWD was investigated by searching for studies that address the following three questions: 3.1.1. Does consuming or omitting breakfast/morning meal have an effect on glycemic control? 3.1.2. Does eating late at night affect glycemic control? 3.1.3. Does meal frequency effect glycemic control?

#### 3.1.1. Does Consuming or Omitting Breakfast/Morning Meal have an Effect on Glycemic control in PWD?

Breakfast skipping among children and adults has increased in the past several decades in the U.S. and worldwide [16,17,18,19,20,21]. Three studies addressed this topic [25,28,29]. All three studies had consistent findings that breakfast consumption might have beneficial effects on glycemic control.

Breakfast consumption was reported to have beneficial effects for patients with T2D. Two cross-sectional studies [25,28] including 31,722 Japanese (mean age = 54.7 years) and 194 American patients (mean age = 59.5 for breakfast eaters and 49.6 years for breakfast skippers), respectively, showed that breakfast skippers had higher body mass index (BMI) and HbA1c values compared to breakfast consumers. A randomized controlled trial [29] including 22 patients with T2D (mean age = 56.9 years) recruited from Venezuela revealed a significantly higher glycemic response in breakfast skippers and impaired insulin secretion manifesting as lower plasma insulin levels after lunch and dinner compared to those who consumed breakfast.

The relationship between breakfast consumption/omission and these health effects may be explained by several potential mechanisms. Breakfast consumption ends a prolonged overnight fasting period and may be particularly satiating compared with meals at other times of the day [30]. Biologically, breakfast consumption may hinder high concentrations of ghrelin and low concentrations of insulin, which induce hunger and eating [30]. These hormonal regulatory effects of breakfast consumption may translate to reduced appetite, lower caloric intake later throughout the day, improved weight control and lower T2D risk [28,30]; effects may be amplified when breakfast is skipped over 3 days per week compared to fewer days [31]. Additionally, consuming breakfast prior to lunch results in reduced glycemia and enhanced insulin after lunch, or “second meal phenomenon”, due to enhanced beta-cell responsiveness at the second meal induced by the first meal [29]. Lastly, the macronutrient content of breakfast meals, specifically a high fiber content, may be related to improved glycemic response and insulin sensitivity [30]. Taken together, these potential mechanisms may contribute to the beneficial effects of breakfast consumption on glycemic control.

#### 3.1.2. Does Eating Late at Night Affect Glycemic Control in PWD?

Some studies suggest that late meal consumption may have a detrimental effect on glycemic control [32,33,34,35,36]. Three studies addressed this topic [33,34,35]. Findings revealed a consistent association between late dinner/snack consumption and higher risk of obesity, higher BMI, poor glycemic control, hyperglycemia, and higher risk of T2D.

Two cross-sectional studies in a Japanese population [34,35] (*n* = 409, mean 68.3 years and *n* = 478, 20–40 years, respectively) reported significantly higher HbA1c and suboptimal glycemic control in those who ate later during the day (beyond 8:00 p.m. and 10:00 p.m., respectively) compared to those who ate earlier. Another randomized crossover trial [33] including 16 Japanese patients with T2D (mean age = 70.3 years) reported higher mean plasma glucose and incremental area under the curve (iAUC) for glucose, an index of glucose excursion, among those who ate a later dinner (9:00 p.m.) compared to those who had an early dinner (6:00 p.m.).

The negative outcomes associated with eating later in the day (decreased glucose tolerance and increased insulin resistance) may be explained by misalignment between circadian rhythm and hormonal regulation [32] as well as prolonged PP glucose spikes [34] caused by late meal consumption. Additionally, late dinner, commonly accompanied with overeating, might lead to poor glycemic control [34]. Evidence suggests that late eating may have a negative effect on glycemic control.

#### 3.1.3. Does Meal Frequency, or Number of Times Eating Events Occurred throughout a Day, Effect Glycemic Control in PWD?

Eating frequency and/or snacking, as in the number of times eating events occurred throughout a day, is an additional factor with potentially profound impacts on health [37] and disease risk through links with energy intake, body weight and metabolic biomarkers [27,38]. Studies addressing this topic numbered two [37,39], providing consistent findings on optimal meal frequency to manage diabetes in patients with T2D [37,39]. Both studies showed that infrequent meal consumption has a positive effect on glycemic control [37,39].

The two randomized crossover studies included patients with T2D [37,39] in the Czech Republic (*n* = 54, age 30–70 years) and Sweden (*n* = 19, mean age = 63 years). In both studies, authors reported better glycemic control with fewer larger meals compared to more frequent smaller meals. Overall, current evidence suggests infrequent meal consumption might be associated with improved glycemic control among patients with T2D.

A potential reason for the increased T2D risk associated with higher meal frequency compared to lower meal frequency may be related to an increase in total caloric intake throughout the day due to overeating, consuming energy-dense foods, especially those high in refined carbohydrates, increased portion size, and decreased quality of foods [31,37]. Furthermore, large amount of nutrients in infrequent meals, such as fat and protein, could be responsible for a prominent increase of glucose-dependent insulinotropic polypeptide level, inducing enough increase in insulin after the meal to maintain a controlled glucose excursion [39].

In summary, time of eating is a potentially important factor to consider in developing an image-based dietary assessment tool designed for optimizing metabolic control. Specifically, breakfast consumption/skipping, late eating, and frequency of meals may be relevant factors to glycemic control. Evidence shows that breakfast skipping is related to increased risk of obesity, insulin insensitivity, and higher risk of T2D compared with eating breakfast [25,26,29]. Findings on late eating/snacking behaviors showed negative effects on glycemic control, which could be due to the impact of late eating on energy intake and PP blood glucose response [34]. Meal frequency might be associated with neurological and hormonal responses that affect blood insulin and glucose levels and infrequent meal consumption was associated with improved glycemic control [37,39]. Quantifying the timing of dietary intake is promising to inform the development of a sensitive and targeted dietary analysis system for the management of diabetes and should be considered in a MIDAT. In addition, MIDAT capability to send mobile notifications to prompt and change eating behaviors in terms of time of eating could help improve glycemic control.

### 3.2. Determining How the Macronutrient Composition of Meals Effects Glycemic Control in PWD

Determining the appropriate mealtime insulin dose, especially for T1D, has mainly focused on the carbohydrate content of the meal, since it is the macronutrient with the greatest effect on PP blood glucose level [40,41,42,43,44,45]. However, recent studies have suggested that other macronutrients including fat, protein [40,41,45] and fiber [46,47] as well as the glycemic index (GI) of foods [48,49,50,51] might also impact PP glucose levels. Furthermore, the order in which major macronutrients are consumed may affect insulin and glycemic response [46,52,53,54]. Knowledge of how the macronutrient breakdown of a meal affects PP glucose response may inform the development of a MIDAT. The potential to capture and identify food through the mobile tool and automatically determine macronutrient information of the meal minimizes the traditional burden of counting carbohydrates and provides automated measures of these components for the entire meal. Furthermore, enhanced macronutrient profiles, with all macronutrients in addition to carbohydrates taken into consideration, may more accurately inform prandial insulin dosing and improve management of PP glucose response compared with traditional carbohydrate counting [40,41,45,46,47]. Therefore, the following questions were addressed in the current goal: 3.2.1. What is the impact of dietary fat and protein on PP glucose levels? 3.2.2. What is the impact of fiber on PP glucose levels? 3.2.3. How does the glycemic index of food/meal effect glycemic response? 3.2.4. Does food order within a meal have an effect on PP glycemic response?

#### 3.2.1. What Is the Impact of Dietary Fat and Protein on PP Glucose Levels in PWD?

Dietary fat and protein delay gastric emptying [42], sustain glycemic excursions [44], impair insulin sensitivity [42], affect glucose metabolism [43], and potentially increase PP glucose levels following their intake. A total of seven studies addressed this question all including patients with T1D [40,41,42,43,44,45,55], with five studies focusing on dietary fat [40,41,42,45,55] and four on dietary protein [40,41,43,44]. Findings from the studies were inconsistent regarding the impact of dietary fat and protein on PP glucose levels, with four studies reporting increased PP glucose levels after consumption of a high fat or protein meal [40,41,42,44] and three studies reporting no effect of meals high in fat or protein on PP glucose concentrations compared to controls [43,45,55].

One randomized crossover trial conducted in Germany [40] (*n* = 15, mean age = 16.8 years) provided two meals: control and high fat meal (twice the amount of fat in the control meal) and another in Australia [41] (*n* = 33, age 8–17 years), provided meals with 4 g vs. 35 g of fat in the control and high fat meals, respectively, and showed delayed peak glucose concentrations and significantly higher PP glucose concentrations 12 h [40] and 180–300 min (3–5 h) [41] after the meal, in participants who consumed meals higher in fat compared to controls, respectively. Furthermore, one randomized crossover trial in the U.S. [42] including 7 participants (age 43–67 years) demonstrated that consuming an extra 50 g of fat in the dinner meal was associated with more hyperglycemia (measured by AUC) during two 18 h continuous monitoring periods and led to an increased insulin requirement, compared to a lower fat meal (10 g), even when additional insulin was administered. Significant differences among individual glucose responses to a high fat diet versus a low-fat diet were also observed [42]. On the other hand, a crossover trial in Sweden including seven adolescents [55] (15.5 to 17.5 years) found larger plasma glucose AUC in participants consuming a low-fat diet (2 g) compared to those consuming a high-fat diet (38 g) two hours post-meal but no difference in time-to-peak of glucose concentration between low-fat and high fat meals. Similarly, a Spanish study [45] with 17 participants (mean age 35.8 years) reported varying glycemic response following up to 180 min (3 h) after consumption of two meals of different fat contents (8.9 g vs. 37.4 g), but glucose levels all remained within the PP targets (70–180 mg/dL).

Dietary protein consumption was evaluated in two crossover randomized trials conducted in Germany (*n* = 15, mean age 16.8 years, two meals: 28 g vs. 110 g of protein in the control and high protein meals, respectively) [40] and Australia (*n* = 33, age 8–17 years, two meals: 5 g vs. 40 g of protein in the control and high protein meals, respectively) [41]. Authors reported higher PP glucose concentration 12 h [41] and 180 to 300 min (3–5 h) postmeal, in participants who consumed meals higher in protein compared to controls, with one study [40] suggesting potential additive effects of fat and protein on blood glucose level (measured by AUC and glucose concentration). Independent evaluation of protein content (without carbohydrates or fats) was completed in one randomized crossover trial in Australia [44] (*n* = 32, age 10–33.4 years) and showed that a high protein diet (75–100 g) resulted in initial lower glycemic excursions (60–120 min (1–2 h) after a meal) then higher glycemic excursions (180 to 300 min (3–5 h) after a meal) compared to excursions following consumption of a lower protein diet (12.5–50 g). Moreover, addition of protein (75–100 g) to a meal led to delayed and sustained glucose peak compared with addition of carbohydrate (20 g) [44]. However, a randomized crossover trial in France [43] (*n* = 28, age 28–47 years) reported no difference in PP glucose level or AUC 2 h after a meal in interstitial glucose and capillary glucose for meals containing different protein amounts.

Several potential mechanisms might explain the findings of the effect of dietary fat and protein on glycemic control among patients with T1D. Delayed hyperglycemia from fat may result from delaying gastric emptying [41,42,45,55]. Additionally, free fatty acids might induce insulin resistance and increase hepatic glucose output causing delayed and increased glucose response [41]. A reduced glucose concentration in the early PP period might be explained by increased glucagon-like peptide-1 secretion (GLP-1, a hormone that decreases blood glucose concentration) induced by a high-fat meal compared to a low-fat meal [55]. On the other hand, high protein in the diet could delay glucose response and potentially result in hyperglycemia by affecting gluconeogenesis, increasing glucagon secretion and impairing insulin’s ability to suppress endogenous glucose production [1,43,44]. Moreover, protein-rich meals (without carbohydrates) may result in a delayed and sustained glucose peak compared to meals with carbohydrates because of a more rapid effect of carbohydrates on PP glucose response compared to protein [44]. Lastly, the type of macronutrients, such as the chain length of fatty acids and the structures of protein, and the physical property of food matrix structure, which is the 3D structure in which macronutrients interact and assemble within the food, could all affect digestion of foods. These complex properties of macronutrients could thus influence digestion and glycemic responses in various ways [56,57,58]. In summary, fat and protein content of a meal might impact PP blood glucose and should be captured by assessment tools and integrated within the calculation of appropriate insulin dosing strategies.

#### 3.2.2. What Is the Impact of Fiber on PP Glucose Levels in PWD?

Diets high in fiber have a protective effect on all-cause mortality in PWD [51]. Specifically, evidence suggests that dietary fiber might be associated with reduced PP glucose and insulin levels due to delayed gastric emptying from added viscosity [48]. Nine studies addressed this topic [47,48,51,59,60,61,62,63,64], with five including patients with T1D [48,59,60,61,62], four including patients with T2D [47,51,63,64], and one involving healthy individuals [64]. Findings regarding the effect of dietary fiber consumption on PP blood glucose were generally consistent, with seven studies showing positive effects in patients with T1D [9,60,62] and T2D [7,51,63,64], and two studies showing no effect [48,61] in patients with T1D.

A randomized controlled trial in Austria [59] (*n* = 38, mean age = 10.8 years) examined the effect of a β-glucan-enriched (a fiber source) bedtime snack (provided at 10:00 p.m.) on nighttime blood glucose levels in patients with T1D and determined lower blood glucose levels until 2:00 a.m. with β-glucan-enriched snacks compared to conventional snacks. Yet, there were no differences in the prevalence of nocturnal hypoglycemia. In a crossover study in the U.S. including patients with T1D (*n* = 10; mean age = 11.2 years) [60], Nader et al. reported no differences in PP mean blood glucose excursions and incidence of hypoglycemia with and without fiber supplementation, although the authors found a strong negative correlation between the amount of fiber supplemented and the mean maximum PP blood glucose after lunch and breakfast. Similarly, a randomized controlled trial in Italy including patients with T1D (*n* = 54, mean age = 28 years) [62] lasting for 24 weeks found that a high-fiber diet improved glycemic control by decreasing mean daily blood glucose concentrations and the number of hypoglycemic episodes compared to a low-fiber diet. On the other hand, a Swedish double-blind placebo-controlled crossover study in 14 patients with T1D (mean age = 47 years) [61] indicated no benefits of pre-meal β-glucan consumption on glycemic control. Another randomized crossover study [48] including nine Canadian patients with T1D (no age reported, mean duration of diabetes was 15 years) reported a small but significant decrease in capillary blood glucose after a high fiber diet (at least 40 g of fiber) compared to control.

Studies including patients with T2D suggested that fiber might be associated with improved glycemic control. One randomized crossover trial in Brazil [63] (*n* = 14; mean age = 68.5 years) reported lower glycemic control among participants who consumed a high glycemic index low fiber diet. Another randomized crossover trial in Brazil [51] (*n* = 19, mean age = 65.8 years) associated higher dietary fiber intake with lower PP blood glucose regardless of the source (food or supplement). Results of an intervention study [47] with 22 Italian participants with T2D (mean age = 68.3 years) showed that consumption of β-glucan rich bread reduced HbA1c, decreased PP and mean blood glucose compared to controls who consumed regular white bread. Similarly, a randomized open-label trial [65] in Korea [64] including 15 healthy individuals (mean age = 47.3 years) and 15 patients with T2D (mean age = 62.9 years) reported lower iAUC for plasma glucose both in healthy participants and those with T2D after consuming a premeal protein-enriched dietary fiber bar compared with ingesting the same bar postmeal, indicating decreased PP glucose excursion.

Several mechanisms might explain the possible beneficial effects of fiber on glycemic control. Fiber increases viscosity of the stomach content leading to delayed gastric emptying and absorption of carbohydrates [47,48,51,61,62,64]. Fiber could also potentially reduce digestion and absorption of carbohydrates by reducing accessibility to digestive enzymes such as α-amylase [51,61,62]. Additionally, the breakdown of fiber could stimulate insulin secretion and improve insulin sensitivity, decreasing blood glucose levels [51,60]. Lastly, fiber could improve satiety and suppress appetite due to “bulking action” [47]. Altogether, these mechanisms may contribute to the potential beneficial effect of fiber consumption in diabetes management.

#### 3.2.3. How Does the Glycemic Index of Food/Meal Effect Glycemic Response in PWD?

Glycemic index, calculated as the iAUC of blood glucose after consuming a test food divided by that of a reference food [66] is used to assess effects of food on PP glucose response [49,50,67]. Evidence from previous studies associated low glycemic index food consumption with improved glycemic response, specifically fasting glucose and HbA1c [49,50,67]. Due to the substantial amount of literature addressing this topic, three review articles were examined to answer this question [49,50,67]. Results of all three review articles showed that glycemic index is an important consideration for diabetes management and might have beneficial effects on glycemic control.

Ojo et al. [50] conducted a meta-analysis including 6 randomized controlled trials in patients with T2D; compared to a control group who consumed a high-glycemic index diet, those who consumed a low-glycemic index diet had improved HbA1c (in two studies), fasting blood glucose (in four studies), but showed no differences in PP glycemic response (in four studies). Another review including 54 randomized controlled trials [67] examined whether low glycemic index diets, compared with other diets (such as high-GI, low-fat, low-carbohydrate, conventional weight loss diets or specialty diets), lead to lower measures of blood glucose control in patients with T1D, T2D or those with glucose intolerance. The authors [67] reported that low glycemic index diets significantly reduced HbA1c (36 studies) and fasting blood glucose (46 studies) compared to other diets, with a greater reduction of fasting blood glucose in patients with T2D compared to those with T1D or glucose intolerance. Similarly, a review by Wang et al. [49] investigated the effects of low glycemic index food on HbA1c in patients with T1D and T2D. The authors reported lower HbA1c levels with low glycemic index food compared to high-glycemic index food in T1D and T2D patients.

One potential explanation for the beneficial effects of a low glycemic index diet on glycemic measurements might be related to the fiber content of foods; since low glycemic index foods are usually high in fiber [50] it is difficult to ascertain whether this beneficial effect is due to glycemic index of a meal, its fiber content, or a combination of both. High fiber supports a gradual supply of glucose to the bloodstream which results in a lower and more stable release of insulin and better glycemic control compared with low fiber diet in PWD [49].

#### 3.2.4. Does Macronutrient Order within a Meal Have an Effect on PP Glycemic Response in Healthy Individuals and PWD?

The order of how various macronutrients are consumed during a meal, such as preloading protein, fat or fiber-rich food may influence PP glucose level, glucose excursion and plasma insulin by slowing gastric emptying and stimulating secretion of digestive hormones and insulin before the main load of carbohydrates [46,51,53,68]. Studies addressing this topic numbered five [9,46,52,53,68], with four including patients with T2D [46,52,53,68], one including those with T1D [9] and one involving healthy adults as controls [53]. The reviewed articles present consistent findings and suggest that a carbohydrate-last food order might be effective for improving PP glycemia.

A crossover study conducted in the U.S. [52] including 11 patients with T2D (mean age = 54 years) showed that mean PP glucose levels, iAUC for PP glucose, PP insulin levels and iAUC for insulin were significantly lower when non-starchy vegetables (lettuce and tomato salad with low-fat dressing and steamed broccoli with butter) and protein were consumed first, before carbohydrates, compared with the reverse food order. Similarly, another crossover study in patients with T2D in the U.S. [46] (*n* = 16, mean age = 57.7 years) showed lower iAUC for PP glucose, incremental glucose peaks, and PP insulin excursion and higher GLP-1 excursion in a carbohydrate-last meal pattern compared to a carbohydrate first pattern. Moreover, one randomized crossover trial in Japan [53] including 12 patients with T2D and 10 healthy adults (30 to 75 years) demonstrated that compared to consuming rice before fish, consuming fish/meat before rice lead to decreased PP glucose excursions, increased GLP-1 secretion and delayed gastric emptying in both healthy participants and those with T2D. Another study including patients with T2D in Australia [68] (*n* = 8, mean age = 58 years) examined the effects of three different meal orders: whey preload before carbohydrate meal, whey with carbohydrate meal, or no whey given in the meal. The authors reported slowest gastric emptying and greatest GLP-1 after a whey preload and significant lower iAUC for glucose after the whey preload and whey in the meal compared to the no whey meal. One crossover study including 20 patients with T1D [9] (age 7–17 years), showed that consumption of protein and fat prior to carbohydrates resulted in lower PP glucose levels, lower glucose excursions, and a reduction in the total time period in hyperglycemia compared to consuming all macronutrients together. In summary, the reviewed evidence suggests that a carbohydrate-last meal pattern might have potential in improving PP glucose response in patients with T1D and T2D, therefore macronutrient order should be considered to inform insulin dosing and diabetes management.

One mechanism explaining the potential beneficial effect of carbohydrate-last food order on PP glucose concentration might be that pre-consumption of protein and/or vegetables delays gastric emptying and ultimately lowers the rate of carbohydrate absorption and blood glucose levels [46,53,68]. The slowed gastric emptying may also stimulate various hormones such as insulin, incretin (hormone that stimulates decrease in blood level), GLP-1 and cholecystokinin (hormone that stimulates digestion of fat and protein) [46,68]. Moreover, dietary carbohydrates consumed after vegetables might require less insulin due to the subsequent effects on blood glucose level by the dietary fiber in the vegetables [53].

In summary, there is evidence that macronutrient components including fat, protein, and dietary fiber as well as other factors such as the glycemic index of a food and food order may impact glycemic control and thus should be considered and further investigated to determine optimal insulin dosing to improve diabetes management. Some but not all evidence suggests that dietary fat and protein could delay and increase glycemic response, therefore increasing insulin needs [40,41,42,44]. There are inconsistent findings regarding the effect of dietary fiber intake on glycemic control in patients with T1D [48,59,60,61,62], while some evidence shows fiber might be beneficial for PP blood glucose parameters among those with T2D [47,51,63,64]. Reviewed studies show that carbohydrate-last food order may improve PP glucose profile [9,46,52,53,68]. Additionally, studies investigating the link between glycemic index and glycemic response provided evidence suggesting an improved glycemic response, specifically fasting blood glucose and HbA1c, associated with consumption of low glycemic index foods [49,50,67]. Nevertheless, evidence has shown significant interpersonal variations in glucose response by the various dietary components, indicating a need for individualized nutrition therapy for diabetes management. Understanding how the macronutrient breakdown of the meal and other factors including glycemic index and food order affect glycemic response is essential for the development of a MIDAT that can capture these components and link information to insulin dosing requirements to improve diabetes management and control.

## 4. Discussion

The reviewed evidence suggests that time of eating, macronutrient composition and macronutrient order of the meal are important factors to explore and incorporate in the development of a novel MIDAT with the ultimate goal of personalized diabetes management. MIDATs might be able to accurately record the time of meal/snack consumption and capture the quantity and composition of the food consumed. To achieve this, MITDAT could analyze the photo taken by the users and ask follow-up questions to confirm the correct identification of the foods. MIDATs also provide new possibilities in diabetes management. For example, the potential exists for generating a score or indicator for the risk of suboptimal glycemic control based on algorithms developed by incorporating users’ dietary information. This score would be reviewed by health care professionals and could be used as a reference to inform adjustments on medical treatments (diabetes medications and insulin dosing requirements) for PWD. The dietary information collected and analyzed by the MIDAT may be sent in real-time to healthcare professionals for review, consultations, and recommendations. This dietary assessment tool could also lower the burden of diabetes management by providing a practical and inexpensive way to perform remote/virtual medical appointments by collecting dietary information, communicating with practitioners, and intervening to improve outcomes especially for those of low-income or with limited access to healthcare facilities. Research has found that similar MIDAT for patients with cystic fibrosis was effective in supporting medical monitoring, adjusting medication and improving quality of life in patients. [69,70,71,72]. Other possibilities of this novel tool include, but are not limited to, analyzing dietary inputs to determine insulin requirements, linking and incorporating information from continuous glucose monitors, adjusting insulin dosing and completing delivery through insulin pumps to create a self-regulating, real-time responsive, “smart” system for PWD. Dosing strategies incorporating macronutrient information will also need to be further developed to fully utilize this detailed dietary information [13,14,15,73]. Future advancement of image identification technologies might also allow the identification of different types of dietary fats and fiber contained in the meal consumed and incorporate into the dietary information to inform insulin dosing and delivery adjustments. Individualized diabetes management and personalized insulin therapy based on collected dietary information and real-time communication with healthcare professionals, may be achieved by utilizing this novel “smart” technology.

There are several limitations of the evidence in the current review. For instance, cross-sectional study designs [25,28,34,35,38] cannot infer causation. Additionally, included randomized-controlled trials were short in duration which may not reflect the long-term health effects of the interventions [61], and had small sample sizes [51,61,68] limiting the representativeness of the findings. The lack of objective measurements for compliance could lead to self-reporting bias in dietary assessment due to social desirability or failure to remember details or estimate intake. Furthermore, premeal glucose levels, total calories, and calories consumed the day before which could all be related to PP glucose responses were not measured or controlled for in most studies. Also, heterogeneity of test meals (different macronutrient components and glycemic index), differences in PP glucose monitoring periods and measures of glycemic control (HbA1c, fasting and PP blood glucose levels, glucose AUC and insulin AUC) preclude comparisons of findings across the included studies. Lastly, the current review did not include evidence of physical activity, which is an important factor in metabolic control and diabetes management. Admittedly, the MIDAT might not be able to collect physical activity data on its own, it should be easy for the MIDAT to connect and collaborate with another tool, such as another app or device, to gain access to such data and utilize them for informing diabetes management, including insulin dosing and other medical treatments.

Both dietary modification and medical intervention are important components in diabetes management. A MIDAT tailored for better glycemic control in diabetes management should be able to capture and analyze dietary behaviors and intake information and allow communication between healthcare providers and patients.

## 5. Conclusions

Time of eating, macronutrient content and macronutrient order of meals are important aspects to be captured in a MIDAT to enable personalized, real-time diabetes management.

## Data Availability

Not applicable.

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
