# Peer review of "Dietary Aspects to Incorporate in the Creation of a Mobile Image-Based Dietary Assessment Tool to Manage and Improve Diabetes"

_nutrients, 2021, doi:10.3390/nu13041179_

Round 1
Reviewer 1 Report
The objective of this review is to summarize papers that evaluates the role of time of eating occasions on biomarkers of glycemic control
I recommend:
Line 36; 34.2 million people of all ages have diabetes including Type 1 (T1D, an autoimmune disease where the pancreas produces little or no insulin) and Type 2 (T2D, the most common form of diabetes where the body does not use insulin properly and cannot keep blood glucose at normal levels) [1], making diabetes the seventh leading cause of death in the United States (US) [2]
line 51: Add some information about "Insulin shots are most effective when you take them so that insulin goes to work when glucose from your food starts to enter your blood. For example, regular insulin works best if you take it 30 minutes before you eat." ultra-rapid-acting” insulin cloud reduce this time
Line 66: Add information reggarding day-time patterns of carbohydrate intake; https://www.ncbi.nlm.nih.gov/pmc/articles/PMC6835827/
line 85: hemoglobin A1c (HbA1c, an indicator of blood glucose 85
control over the past three months) glycated hemoglobin (HbA1c)
Materials and methods section should be improved including how many papers were excluded and the main reasons. A figure can help in this issue.
Line 127: number of health outcomes, especially metabolic health
(blood glucose, blood lipid, blood pressure, etc.) cardiovascular biomarkers
Line 138, 167, 186, 237, 302, 354 : In Italic please
Line 160: All this effects may be especially increased when breakfast is skipped on ≥3 days per week https://www.ncbi.nlm.nih.gov/pmc/articles/PMC7832891/
line 205: metabolic control
line 216: I think that maybe adding some mobile notifications can help in this issue
line 451 and 498: please add a macronutrient order
line 356: delete glucose or white bread. In addition, the role of Mediterranean Diet should be included.
line 479: add some limitation about the role of physical activity and this correlation with metabolic control and MIDAT
Reviewer 2 Report
The authors address a very relevant topic and present an excellent review about the factors affecting the response against nutrient intake and eating behaviors in patients with diabetes. The paper reads very well, the amount and quality of the information is high, and the potential impact of the presented information in developing future self-management and monitoring tools to support the treatment of these patients supports the publication of this work.
There are some comments that could improve the manuscript:
Title: avoid using hyphens
Abstract: “results showed (…) may affect glycaemic control” please briefly cite some of the effects of each question: nutrient intake and eating behavior.
INTRODUCTION
L66: studies in other fields have also shown and explained the mechanism of interactions among macronutrients (at molecular level, in the context of in vitro simulation of digestion) in foods. This could be mentioned.
Reference
Calvo‐Lerma, J., Fornés‐Ferrer, V., Heredia, A., & Andrés, A. (2018). In vitro digestion of lipids in real foods: influence of lipid organization within the food matrix and interactions with nonlipid components. Journal of food science, 83(10), 2629-2637.
L84-87: did mobile-based interventions assessing dietary intake improve study outcomes just because of food recording? If not, please explain how these tools contributed to improvement
L98-101: very pertinent reflection. Would you say that, despite lots of information is captured with the MIDAT, a kind of model or algorithm should be able to “weight” the effect of different inputs (time of eating, macronutrient composition…) to deliver an “optimal dose of insulin”?
RESULTS
The authors have very appropriately stated the content under research questions.
3.1. In this section, there are three sub-headings. While the information in 3.1.1 and 3.1.2 can be easily translated into practical recommendations for PWD, do authors agree that information in 3.1.3. would require more detailed interpretation? For example, when assessing frequency of intake, time is a very relevant variable, as it could interact with other considerations including the energy and physical characteristics of the foods consumed in each eating event, which could have an impact in gastric emptying, transit time, etc., possibly leading to overlapping between intakes…
Related to “frequency” of intake, are there studies assessing “regularity” of intake? Is there any study assessing the effect of irregular eating times in different days?
3.2. To be coherent with 3.1. better to write “determining”
L228-232: is there any evidence of the underpinning mechanisms explaining these findings?
L238-242: the reported inconsistent results might be attributed to the fact that not only macronutrient composition or proportion is relevant during digestion and absorption, but also the type of macronutrient (e.g. long chain fatty acid triglycerides, the stereospecific position of the fatty acids in the triglyceride molecule, or globular or fibrillar protein structure), and more importantly, the physical properties of the food matrix structure. Commenting on this would enrich the results. This references would be indicated:
Calvo-Lerma, J., Asensio-Grau, A., Heredia, A., & Andrés, A. (2020). Lessons learnt from MyCyFAPP Project: Effect of cystic fibrosis factors and inherent-to-food properties on lipid digestion in foods. Food Research International, 133, 109198.
Asensio-Grau, A., Calvo-Lerma, J., Heredia, A., & Andrés, A. (2021). In vitro digestion of salmon: Influence of processing and intestinal conditions on macronutrients digestibility. Food Chemistry, 342, 128387.
Asensio-Grau, A., Peinado, I., Heredia, A., & Andrés, A. (2018). Effect of cooking methods and intestinal conditions on lipolysis, proteolysis and xanthophylls bioaccessibility of eggs. Journal of functional foods, 46, 579-586.
L463-467: as this review aims at being used as a support to develop MIDAT, previous studies with the same purpose could be cited to reinforce the validity of the purpose. Indeed, a mobile app with dietary intake recording (even with no advanced technologies such as image scanning) has recently shown to be effective in supporting close medical follow-up and improved gastrointestinal-related quality of life in patients with another pathology (Cystic Fibrosis). Also the cited app was able to adjust the dose of medication (pancreatic enzymes) that patients have to match in each meal according to the characteristics of food (mainly amount of fat but also macronutrient composition and other food matrix properties). The reviewer sees a parallelism between the referred study and the scope of the present review. So, the authors could support the hypothesis in these lines with this example. The supporting refrences would be:
Floch, J., Vilarinho, T., Zettl, A., Ibanez-Sanchez, G., Calvo-Lerma, J., Stav, E., ... & Montón, J. L. B. (2020). Users’ Experiences of a Mobile Health Self-Management Approach for the Treatment of Cystic Fibrosis: Mixed Methods Study. JMIR mHealth and uHealth, 8(7), e15896.
Boon, M., Claes, I., Havermans, T., Fornés-Ferrer, V., Calvo-Lerma, J., Asseiceira, I., ... & MyCyFAPP consortium. (2019). Assessing gastro-intestinal related quality of life in cystic fibrosis: Validation of PedsQL GI in children and their parents. PloS one, 14(12), e0225004.
Calvo-Lerma, J., Boon, M., Colombo, C., de Koning, B., Asseiceira, I., Garriga, M., ... & Ribes-Koninckx, C. (2020). Clinical evaluation of an evidence-based method based on food characteristics to adjust pancreatic enzyme supplements dose in cystic fibrosis. Journal of Cystic Fibrosis.
Calvo-Lerma, J., Martinez-Jimenez, C. P., Lázaro-Ramos, J. P., Andrés, A., Crespo-Escobar, P., Stav, E., ... & Ribes-Koninckx, C. (2017). Innovative approach for self-management and social welfare of children with cystic fibrosis in Europe: development, validation and implementation of an mHealth tool (MyCyFAPP). BMJ open, 7(3).
